

# Modeling the drying process of Masson pine needle fuel beds under different packing ratios based on two-phase models in the laboratory

Yunlin Zhang

School of Biological Sciences, GuiZhou Education University, Guiyang, Asian, China

## ABSTRACT

**Background**. When the moisture content of a fuel bed is higher than the fiber saturation moisture content ($0.35$ g g$^{-1}$), the drying process is controlled by evaporation ($>0.35$ g g$^{-1}$) and diffusion ($>0.35$ g g$^{-1}$). Packing ratio has a significant effect on the drying process. Ignoring the impacts of packing ratio or the separate phases of the drying process is one main reason for inaccurate moisture content predictions.

**Method**. This study simulated the drying process in five Masson pine (*Pinus massoniana* Lamb.) needle beds with different packing ratios. Using the fiber saturation moisture content as the cut-off point, we divided the drying process into two phases. The drying mechanism of each phase was different and had its own drying equation. Using a model that does not distinguish the two phases of the drying process as a comparison, the prediction effect of the two-phase model was analyzed. The influence of the fuel bed packing ratio on the drying process was also analyzed.

**Results**. We found that, regardless of any changes in packing ratio, the two-phase model could better simulate the drying process, with a mean absolute error (MAE) and mean relative error (MRE) of the two-phase model 18.4% and 25.6% less than the one-phase model, respectively. The time-lag prediction model was established with the packing ratio, and the errors were all within the allowable range, but the prediction effect of the time-lag prediction model based on the two-phase model was larger.

**Conclusion**. It was further demonstrated that considering the packing ratio of the fuel bed and distinguishing the two separate phases of the drying process could both effectively improve the prediction accuracy of the moisture content of fuel beds based on the semi-physical method.

Corresponding author
Yunlin Zhang, zhangyun-lin@gznc.edu.cn

## INTRODUCTION

The moisture content of fuel determines the possibility of ignition and a number of fire behaviors after ignition. Improving the prediction accuracy of moisture content calculations of fuels is important to forest fire management (*Molina, Ortega & Rodriguez, 2022*; *Rossa, 2017*; *Saglam et al., 2006*; *Zhang & Tian, 2021*). The current prediction models of fuel moisture content are mainly comprised of physical models, empirical models, and semi-physical models (*Aguado et al., 2007*; *Resco de Dios et al., 2015*; *Sun, Yu & Jin, 2015*).

Among these, the semi-physical model has the stability of the physical model and the simplicity of the empirical model, as it uses the moisture mechanism model as the main equation, with parameters determined by statistical methods. It is currently the most widely used method for predicting the moisture content of fuels (*Ruiz & Vega, 2007*; *Sun, Yu & Jin, 2015*).

Equilibrium moisture content and time lag are two key parameters in the semi-physical model. The semi-physical prediction model relies on these two parameter values being accurately measured in order to accurately predict the moisture content of the fuel bed (*Zhang et al., 2018*). Equilibrium moisture content (EMC) indicates the moisture content at which the fuel bed is neither gaining nor losing moisture when the temperature and humidity remain unchanged (*Viney, 1991*). With changes in temperature and humidity, both the moisture content and EMC of the fuel bed will change, but the change in moisture content lags compared with the change in EMC. This lag is expressed as time lag, which is defined as the time taken for a fuel bed to achieve 63% (1-1/e) of its ultimate change based on the assumption that the moisture content of the fuel bed converges exponentially toward its equilibrium value (*Catchpole et al., 2001*).

Methods for calculating EMC and time lag have been intensively reviewed (*Anderson, 1990*; *Catchpole et al., 2001*). *Simard (1968)* uses the humidity bar as the research object and establishes the prediction model of EMC by statistical analysis. *Nelson (1984)* chose the physical method to establish the prediction model of EMC and time lag, achieving a better extrapolation effect. *Van Wanger (1972)* established the key parameter prediction model of the drying process and adsorption process. Other researchers studied the fuels of Mongolian oak and Korean pine and calculated the EMC and time lag prediction models under different packing ratios of fuel beds (*Jin et al., 2016*; *Zhang et al., 2018*). Although a large number of studies on EMC and time lag have been carried out, most of them only study humidity bars or fuel monomers as research objects and so cannot represent moisture changes in fuel beds with complex structures. In particular, packing ratio has a significant impact on the dynamic change in moisture content (*Matthews, 2006*), affecting the outward diffusion path of water. Calculating the moisture content of the fuel bed without considering the bed structure leads to inaccurate moisture content calculations. Therefore, it is important to analyze the influence of packing ratio on changes in fuel bed moisture and to establish a prediction model of key parameters based on the packing ratio of the fuel bed.

Existing research, including most of the studies cited previously, on the moisture content of fuel beds have focused on the impact of the drying process, especially after rain, to the combustible moisture content. However, these studies also assume that the entire drying process is controlled by only one mechanism and that there is only one time lag and EMC from the post-rain moisture content to the combustible moisture content. However, *Nelson & Hiers (2008)* pointed out that when the moisture content of the fuel bed exceeds the fiber saturated moisture content, which is approximately 0.35 g g$^{-1}$, the water loss is mainly the free water on the surface of fuel and between fuel monomers, which is controlled by evaporation, but when the transfer rate of the internal free water to the needle surface becomes slower than the potential rate at which water can evaporate at the surface under

prevailing conditions, the drying mechanism changes to the combined diffusion of bound water and water vapor. *Jin & Chen (2012)*, studying Scots pine needles, found that the drying process of the fuel bed can be divided into two phases controlled by different mechanisms: first evaporation, then diffusion. If both phases of the drying process are not included, EMC and time lag calculations will be inaccurate, affecting fuel bed moisture content estimates and fire behavior predictions based on those estimates.

Each drying phase should be represented by a different drying model, with its own EMC and time lag. Therefore, it is also important to: (i) understand whether there is a significant change in the key parameters when the drying phase is not distinguished compared to when the drying phase is distinguished, and which method has a better prediction effect; (ii) understand the influence of the packing ratio on key parameters, and how these change when including both phases of the drying process and when not distinguishing the drying process; (iii) build a prediction model for key parameters based on the packing ratio of the fuel bed both using a two-phase drying process and not distinguishing the drying process, and see which prediction model is more accurate.

In an attempt to address these issues, we constructed needle beds of Masson pine (*Pinus massoniana* Lamb.) with different packing ratios indoors, analyzed the drying process, and calculated the EMC and time lag (comparing the results when distinguishing both phases of the drying process and when not distinguishing the drying process). We also analyzed the influence of the packing ratio on key parameters, and established a prediction model of those key parameters. Our results are important for understanding the drying process mechanism in fuel beds and for improving the prediction accuracy of moisture content based on semi-physical models. Masson pine is the main pine plant in the southwest forest area of China. Its needles are rich in oil and ignite easily (*Hu, 2005*). The southwest forest area is the second largest forest in China, with interlaced agriculture and forestry, high mountains, and steep slopes. Once a forest fire starts, it is difficult to stop, posing a serious threat to the safety of local residents (*Zhang, Guo & Hu, 2021*). Based on these factors, we used the needles of Masson pine as the research object of this study.

## MATERIAL & METHODS

### Investigation of fuel bed characteristics and sample collection in the field

A representative Masson pine forest was selected to set a sample plot of 25.82 m × 25.82 m. The basic information of the sample plot is shown in Table 1. The fire prevention period in the study area is from October to May, with February through April considered the period of highest fire risk. Because the drying process of fallen pine needles and weathered pine needles are different (*Anderson, 1990*), and to make the research more applicable to forest fire prediction, weathered Masson pine needles were collected at peak fire risk in March.

### Indoor fuel bed construction with different packing ratios

The packing ratio of the fuel bed is a measure of how tightly compacted the Masson pine needles are in the fuel bed. The higher the packing ratio, the tighter the needles are. Packing ratio is calculated using the following formula: $\beta = \frac{\rho_b}{\rho_p}$, where $\beta$ is the packing ratio; $\rho_b$ is
| Table 1 Sample plot information. | | | | | | | |
|---|---|---|---|---|---|---|---|
| Forest type | Position | Slope | Mean BDH (cm) | Mean height (m) | Canopy density | Mean depth of fuelbed (cm) | Mean packing ratio of fuelbed |
| Masson pine | Down | Flat | 21.3 | 15.6 | 0.76 | 6.90 | 0.027 |

the bulk density of the needle bed, which is calculated by the quality and volume of the bed (kg m$^{-3}$); and $\rho_p$ is the particle density of Masson pine needles, which is a fixed value, obtained from the literature, of 543.6 kg m$^{-3}$ (*Hu, 2005*). To ensure our indoor research samples represented the field, the fuel bed thickness in this study was uniformly set to 6.9 cm, which was the average thickness of the field needle bed, and the packing ratio was set to 5 gradients identified in the field: 0.016, 0.021, 0.027, 0.040 and 0.061.

A topless iron frame with a length of 29 cm and a width of 21 cm was chosen to hold the Masson pine needles, and a cloth was placed on the sidewalls to prevent vapor exchange. The bed volume of the Masson pine needles was $4.2 \times 10^{-3}$ $m^{-3}$ (volume of the fuel bed = length * width * thickness = (29 cm * 21 cm * six cm)/10$^{o6}$). Using the packing ratio gradients set in this study and the particle density ($\rho_p$) of Masson pine needles, we calculated the quality of needles corresponding to each packing ratio, using the equation: $w = \rho_b v = \rho_p \beta v$, where $w$ is the quality of needles corresponding to each packing ratio and $v$ is the volume of the fuel bed.

## Simulating the fuel bed drying process

To more accurately determine the effect of the fuel bed packing ratio on the drying process, we conducted all drying process experiments under one consistent temperature and humidity ratio, excluding the effects of temperature, humidity, solar radiation, wind, and duff moisture code. The temperature and humidity were set at 25 °C and 0.60 g g$^{-1}$, respectively, which represent the average conditions of the high fire risk period in the study area.

The drying process used in this study is outlined as follows: (1) The collected needles were dried in an oven until the weight did not change, and then needles corresponding with the quality of each gradient of packing ratio were soaked in water for 24 h to reach the saturated moisture content *Pan, Yang & Qu, 2002*). (2) The saturated needles were then removed and drained, the free surface water was wiped off, and the weight was recorded, similar to the methods of *Jin & Chen (2012)*. (3) The needles prepared in step (2) were placed in the tray to construct fuel beds with different packing ratios. The needle beds were then placed in a constant temperature and humidity box, with an automatic weighing balance (AS. US) in the box to automatically record the data every 10 min until the weight of the needle bed remained constant. This drying process experiment was repeated three times for each packing ratio, and the arithmetic mean moisture content value of the three experiments was recorded for each packing ratio. The basic outline of the experiment is shown in Table 2.

**Table 2  Summary of fuelbed characteristics and environmental conditions.**

| Number | Fuelbed depth (cm) | Fuelbed packing ratio | Fuel load (g m$^{-2}$) | Initial moisture content (g g$^{-1}$) | Final moisture content (g g$^{-1}$) |
|---|---|---|---|---|---|
| 1 | | 0.016 | 596.06 | 0.972 | 0.176 |
| 2 | | 0.021 | 799.67 | 0.986 | 0.146 |
| 3 | 6.90 | 0.027 | 1029.72 | 0.995 | 0.173 |
| 4 | | 0.040 | 1525.12 | 0.988 | 0.177 |
| 5 | | 0.061 | 2309.85 | 0.993 | 0.155 |

## Model description

Research shows that when the temperature and humidity are constant, for dead fuels with a small Biot number (the ratio of internal moisture diffusion and external convection resistance to water vapor movement), the moisture content of the fuel bed can be calculated, as shown in Formula Eq. (1), (*Byram & Nelson, 1963*; *Viney, 1991*). This formula is especially effective for simulating the drying process of dead fuel in forests (*Jin, Li & Li, 2000*). The EMC and time lag in the drying process under different packing ratios can also be calculated using Formula Eq. (1), hereafter referred to as the one-time lag model:

$$M = E + A * \exp\left(-\frac{t}{\tau}\right) \tag{1}$$

where $M$ is the moisture content of the fuel bed (g g$^{-1}$); $E$ is the equilibrium moisture content of the fuel bed (g g$^{-1}$); $\tau$ is the time lag of the fuel bed (h); and $t$ is the time (h).

A moisture cut-off point distinguishes the transition between the evaporation phase of the drying process and the diffusion phase. Since the two phases of the drying process are controlled by different mechanisms, different models need to be applied to represent the two drying phases. In this study, the moisture content cut-off point was set at 0.35 g g$^{-1}$ (saturated moisture content, *Luke & McArthur, 1978*) for the two drying phases. Based on this, a two-time lag model was developed to describe the drying process in two phases, from the initial moisture content to 0.35 g g$^{-1}$ and from 0.35 g g$^{-1}$ to the EMC, with each phase having an independent set of parameters, including EMC and time lag (*Jin & Chen, 2012*).

For the first phase of the drying process, when the moisture content of the fuel bed is greater than or equal to 0.35 g g$^{-1}$, the formula was written as:

$$M_1 = E_1 + A_1 * \exp\left(-\frac{t}{\tau_1}\right). \tag{2}$$

For the second phase of the drying process, when the moisture content of the fuel bed is less than 0.35 g g$^{-1}$, the formula was written as:

$$M_2 = E_2 + A_2 * \exp\left(-\frac{t}{\tau_2}\right) \tag{3}$$

with the symbols representing the same variables as in Formula Eq. (1), and the subscripts 1 and 2 denoting the first and second phases of the drying process.

## Data analysis

The EMC and time lag of each fuel bed was calculated using the nonlinear least squares regression model based on the moisture content value at 10-minute intervals and formulas Eqs. (1)–(3). A $t$ test was selected to analyze and compare whether there was a significant difference between the results of the two-time lag model and the results of the one-time lag model. K-fold cross-validation was used to calculate the mean absolute error (MAE) and mean relative error (MRE) of the two models (Formulas Eqs. (4)–(5)). The error of the two-time lag model was defined as the sum of the errors of the two phases. The MAE and MRE were then compared between the two methods to determine whether there was a significant difference.

$$MAE = \frac{1}{n} \sum_{i=1}^{n} |m_i - \widehat{m_i}| \tag{4}$$

$$MRE = \frac{1}{n} \sum_{i=1}^{n} \frac{|m_i - \widehat{m_i}|}{m_i} \tag{5}$$

where $m_i$ is the observed value of the fuel bed moisture content (g g$^{-1}$); $\widehat{m_i}$ is the predicted value of the fuel bed moisture content (g g$^{-1}$); and $n$ is the number of samples in the drying process.

One-way ANOVA and multiple comparisons were used to analyze the effect of the fuel bed packing ratio on the EMC and time lag calculated by the two models, and the best-fitting equation was chosen to represent the effect of the packing ratio. The MAE and MRE of the model were calculated, and the accuracy of the prediction model was compared. Taking the observed values of key parameters as the abscissa and the predicted values as the ordinate, we drew a 1:1 graph, comparing the deviation of the fitted line and the 1:1 line to analyze the prediction effect of the model.

All data analyses were performed in Statistica 10.0 (http://www.statsoft.com; StatSoft, Inc., Tulsa, OK, USA) and in R Studio (RStudio Team, 2022).

## RESULTS

### Parameter estimations

The EMC and time lag of each fuel bed were calculated using formulas Eqs. (1)–(3), and the results are shown in Tables 3 and 4. The range of the EMC using the one-time lag model of the fuel beds with different packing ratios was 0.145 to 0.174 g g$^{-1}$, with a mean value of 0.162 g g$^{-1}$; the time lag range was 7.102–25.783 h, with a mean value of 12.500 h. For the two-time lag model, the EMC and time lag in the first phase ranged from 0.268 to 0.289 g g$^{-1}$ and 4.768 to 16.783 h, respectively, and the mean values were 0.280 g g$^{-1}$ and 8.200 h, respectively; the range of the EMC and time lag in the second phase were 0.134 to 0.160 g g$^{-1}$ and 8.325 to 27.657 h, respectively, and the mean values were 0.152 g g$^{-1}$ and 13.836 h, respectively.

Regardless of how the packing ratio of the fuel bed changed, $E_2$ was significantly higher than $E$. When the packing ratio was 0.040, $E$ was significantly higher than $E_1$, but there was no significant difference found in other packing ratios. No significant difference was found between $\tau_2$, $\tau_1$ and $\tau$, regardless of fuel bed packing ratio (Fig. 1).

Table 3 Estimated parameters of the one-time lag model.

| Packing ratio | $E$ (g g$^{-1}$) | s.e. ($E$) | $A$ | s.e. ($A$) | $\tau$ | s.e. ($\tau$) | $R^2$ |
|---|---|---|---|---|---|---|---|
| 0.016 | 0.173 | 0.006 | 0.750 | 0.019 | 7.102 | 3.434 | 0.997 |
| 0.021 | 0.149 | 0.017 | 0.771 | 0.010 | 7.488 | 1.674 | 0.997 |
| 0.027 | 0.174 | 0.014 | 0.760 | 0.024 | 8.732 | 0.578 | 0.997 |
| 0.040 | 0.171 | 0.003 | 0.756 | 0.001 | 13.384 | 1.304 | 0.997 |
| 0.061 | 0.145 | 0.007 | 0.765 | 0.025 | 25.783 | 3.973 | 0.996 |

Table 4 Estimated parameters of the two-time lag model. $E_1, E_2, \tau_1, \tau_2$, equilibrium moisture content and time lag for the first and second phases, respectively; $A_1, A_2$, empirically determined constants; $R_1^2$ and $R_2^2$, coefficients of determination of the first and second phases.

| Packing ratio | $E_1$ (g g$^{-1}$) | s.e. ($E_1$) | $A_1$ | s.e. ($A_1$) | $\tau_1$ | s.e. ($\tau_1$) | $E_2$ (g g$^{-1}$) | s.e. ($E_2$) | $A_2$ | s.e. ($A_2$) | $\tau_2$ | s.e. ($\tau_2$) | $R_1^2$ | $R_2^2$ |
|---|---|---|---|---|---|---|---|---|---|---|---|---|---|---|
| 0.016 | 0.268 | 0.017 | 0.691 | 0.031 | 4.951 | 2.369 | 0.160 | 0.009 | 0.187 | 0.009 | 8.325 | 3.701 | 0.999 | 0.999 |
| 0.021 | 0.273 | 0.018 | 0.697 | 0.021 | 4.768 | 0.827 | 0.145 | 0.018 | 0.203 | 0.018 | 8.717 | 1.966 | 0.999 | 0.999 |
| 0.027 | 0.285 | 0.022 | 0.692 | 0.022 | 5.916 | 0.483 | 0.171 | 0.014 | 0.183 | 0.013 | 9.196 | 0.431 | 0.998 | 0.997 |
| 0.040 | 0.289 | 0.006 | 0.678 | 0.003 | 8.580 | 1.370 | 0.150 | 0.005 | 0.201 | 0.005 | 15.283 | 2.780 | 0.998 | 0.998 |
| 0.061 | 0.285 | 0.028 | 0.670 | 0.040 | 16.783 | 4.076 | 0.134 | 0.013 | 0.218 | 0.012 | 27.657 | 5.209 | 0.995 | 0.999 |

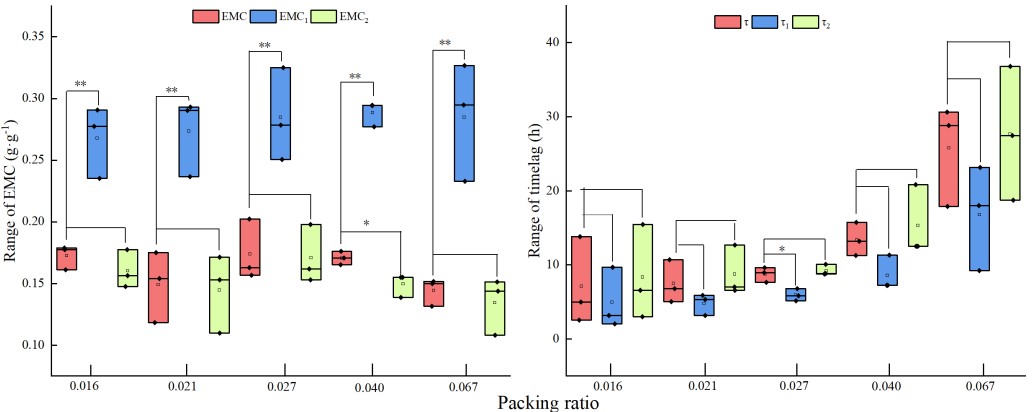

Figure 1 Key parameter test results under different packing ratios. Two asterisks (**) indicate a very significant difference; one asterisk (*) indicates a significant difference.

## Model comparison under different fuel bed packing ratios

Figure 2 shows the prediction errors of the one-time lag model and the two-time lag model at different fuel bed packing ratios. Without considering the packing ratio, the MAEs of the one-time lag model and the two-time lag model were 0.0087 g g$^{-1}$ and 0.0071 g g$^{-1}$, respectively, and the MRE values were 2.38% and 1.77%, respectively. The MAE and MRE values from the two models did not significantly differ between packing ratios, but the two-time lag model results were consistently lower than the MAE and MRE values calculated using the one-time lag model, while the coefficient of variation was higher using the two-time lag model than the one-time lag model.
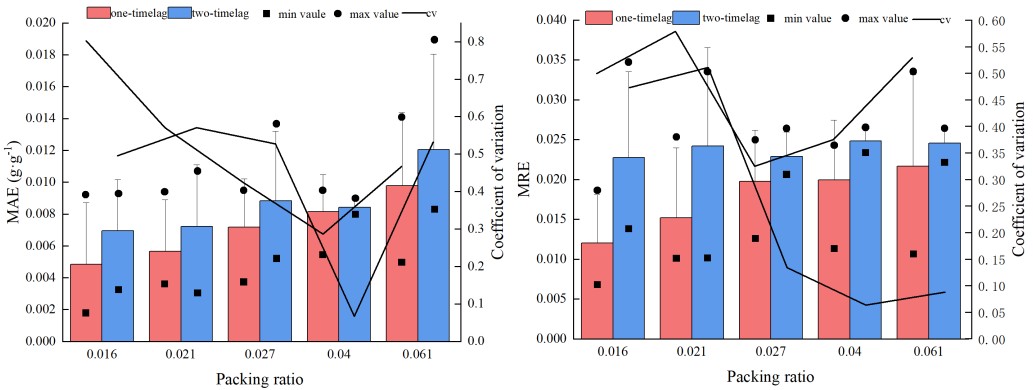

**Figure 2** Error (MAE and MRE) comparison of the two models under different fuelbed packing ratios.

**Table 5** The results of the ANOVA test.

| Model | Index | SS | df | MS | F | P |
|---|---|---|---|---|---|---|
| One-timelag | $E$ | 0.002 | 4 | 0.001 | 1.782 | 0.209 |
| | $\tau$ | 737.059 | 4 | 184.265 | 9.476 | 0.002 |
| Two-timelag | $E_1$ | 0.001 | 4 | 0.000 | 0.208 | 0.928 |
| | $\tau_1$ | 304.101 | 4 | 76.025 | 5.064 | 0.017 |
| | $E_2$ | 0.002 | 4 | 0.001 | 1.245 | 0.353 |
| | $\tau_2$ | 813.632 | 4 | 203.408 | 6.443 | 0.008 |

## Effects of fuel bed packing ratio on EMC and time lag

Table 5 shows the effect of fuel bed packing ratio on the EMC and time lag. The packing ratio of the fuel bed had no significant effect on the EMC in either model, but the time lag in the one-time lag model and the two-time lag model were both significantly affected by the fuel bed packing ratio.

Since the packing ratio of the fuel bed had no effect on the EMC, only the time lag was analyzed below. All time lag calculations increased as the packing ratio of the fuel bed increased, but only when the packing ratio of the fuel bed was 0.061 was this increase significant (Fig. 3).

## Model of time lag with fuel bed packing ratio

According to Fig. 3, when the temperature was 25 °C and the humidity was 0.6 g g$^{-1}$, the time lag increased exponentially with the increase of the fuel bed packing ratio. Therefore, the time lag prediction model based on the packing ratio was: $\tau = a * e^{b * \beta}$ (where $\tau$ is time lag; $\beta$ is the packing ratio; and $a$ and $b$ are parameters to be estimated in the model). The MAE of the prediction model of $\tau$ was 2.174 h, and the MRE was 17.0%; the sums of the MAE and MRE of the prediction model of $\tau_1$ and $\tau_2$ were 0.732 h and 9.69%, respectively (Table 6).

Using the observed value as the abscissa and the predicted value as the ordinate, we drew a 1:1 graph. This showed that the error of prediction model $\tau$ was high, while the error of

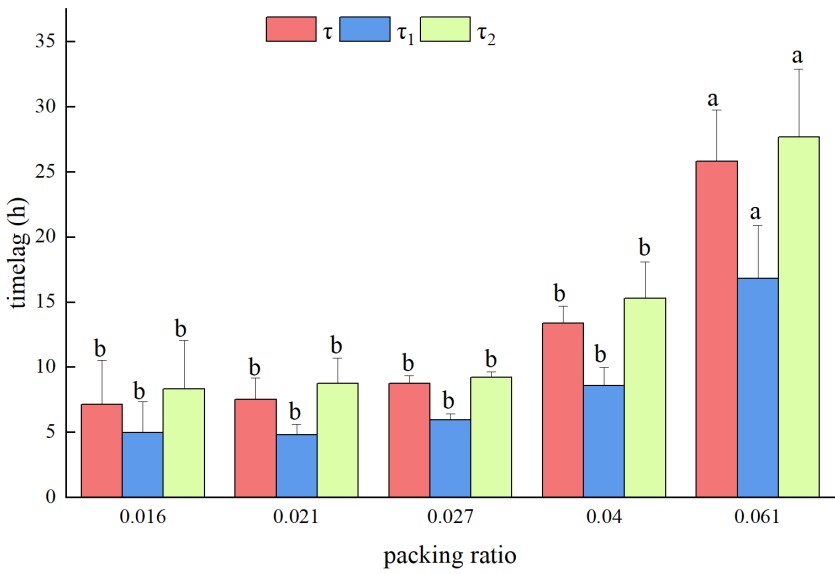

**Figure 3  The change in the time lag with the packing ratio of the fuelbed.**

**Table 6  Time lag prediction model.**

| Prediction model | $R^2$ | MAE (h) | MRE (%) |
|---|---|---|---|
| $\tau = 4.683 * e^{30.570 * \beta}$ | 0.991 | 2.174 | 17.0 |
| $\tau_1 = 2.669 * e^{30.046 * \beta}$ | 0.994 | 0.274 | 4.7 |
| $\tau_2 = 4.705 * e^{29.025 * \beta}$ | 0.993 | 0.458 | 4.9 |

$\tau_1$ and $\tau_2$ were both low. The $\tau_1$ prediction model was the closest to the 1:1 line, followed by $\tau_2$, with $\tau$ being the farthest from the line (Fig. 4).

## DISCUSSION

### Equilibrium moisture content

The key parameters obtained in the temperature and humidity range of this study were similar to those in previous studies. The range of the EMC using the one-time lag model was 0.145 to 0.174 g g$^{-1}$. *Zhang, Sun & Liu (2020)* obtained EMC fuel bed values of Korean pine with different packing ratios ranging from 0.131 to 0.139 g g$^{-1}$. *Bakšić, Bakšić & Jazbec (2017)* obtained an EMC of approximately 15% in Mediterranean pine when the humidity was 0.600 g g$^{-1}$. In our study, when using the two-time lag model, the variation range of the EMC in the first phase was 0.268−0.289 g g$^{-1}$ and 0.134−0.160 g g$^{-1}$ in the second phase. $E_2$ was significantly higher than $E$, but there was no significant difference between $E_1$ and $E$, which was mainly because the EMC was not affected by the initial moisture content but was significantly impacted by the final moisture content (*Jin & Li, 2010*). This study divided the drying process into two phases based on the cut-off point of the fiber saturated moisture content, and the moisture content at the end of the first phase was significantly higher than the moisture content used in the one-time lag model, so $E_2$

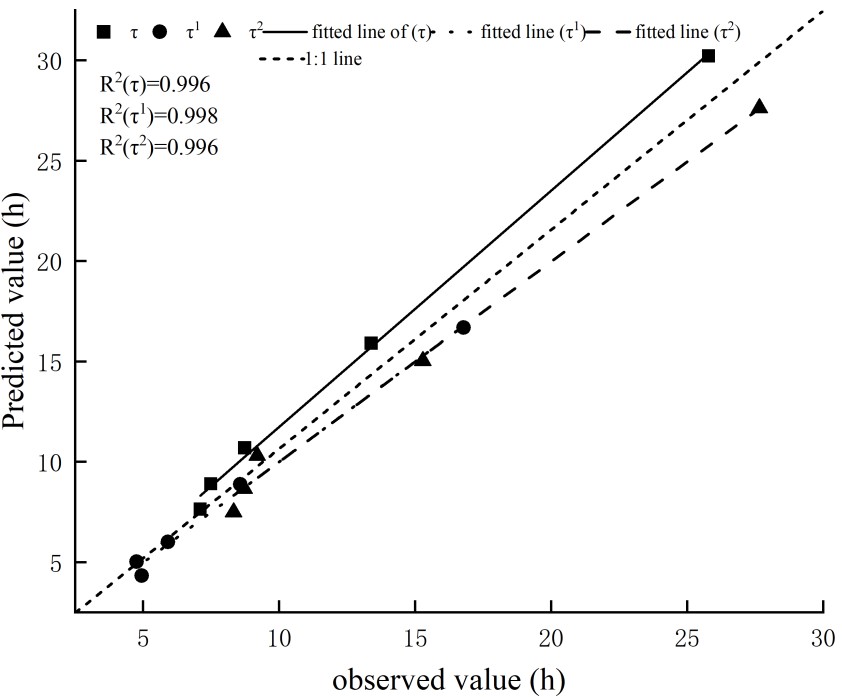

**Figure 4** Comparison of the observed value and predicted value of timelag.

was significantly higher than $E$; the moisture content at the end of the second phase was the same as the moisture content value used in the one-time lag model, so there was no significant difference between $E_1$ and $E$.

## Time lag

The time lag results in this study from both the one-time lag model and the two-time lag model were higher than those found by *Jin & Chen (2012)*, which is mainly due to differences in air temperature and humidity. Studies have shown that the lower the humidity, the faster the fuel bed dries, and the lower the time lag. The humidity in this study was 0.600 g g$^{-1}$, while the humidity range in the Jin and Chen experiment was 0.105−0.342 g g$^{-1}$; the humidity interval in this study was significantly lower than that in Jin's study, so the time lag was higher. The time lag of this study was also higher than that of *Catchpole et al. (2001)*, mainly because their experiments were carried out in the field, and meteorological factors, such as wind speed and solar radiation, accelerated the moisture change in the fuel bed, leading to a reduction in time lag. There was no significant difference between $\tau_1$, $\tau_2$ and $\tau$, but $\tau_1 < \tau < \tau_2$ for different fuel bed packing ratios, which was similar to the research results of previous studies (*Catchpole et al., 2001*; *Jin & Chen, 2012*; *Matthews, 2006*). This is mainly due to the different mechanisms of the drying process. *Pippen (2008)* concluded that the evaporation process was easier than the diffusion process, explaining why $\tau_1$ is the smallest and $\tau_2$ is the largest.

### Analysis of fitting effect under different methods

The MAE values calculated using the one-time lag model and the two-time lag model were 0.0087 g g$^{-1}$ and 0.0071 g g$^{-1}$, respectively, and the MRE values were 2.38% and 1.77%, respectively. Although the errors between the two models were not significantly different, the MAE and MRE of the two-time lag model decreased by 18.4% and 25.6%, respectively, compared to the one-time lag model. In summary, the key parameters obtained by the two models differ, and the calculation results of the two-time lag model were more accurate, meaning the results obtained from the two-time lag model are better than those from the one-time lag model, especially when the moisture content of the fuel bed was low (below 0.25 g g$^{-1}$). Using the two-time lag model will significantly improve the prediction accuracy of fuel bed drying calculations.

### Correlation analysis

Regardless of whether the two phases of the drying process were distinguished, the packing ratio of the fuel bed had no significant effect on the EMC, similar to the findings of *Anderson (1990)*. Packing ratio had a significant effect on $\tau$, $\tau_1$, and $\tau_2$ with the highest effects seen in $\tau$ and $\tau_2$. The drying process of the fuel bed includes both the evaporation of free water and the diffusion of bound water. As the fuel bed packing ratio increases, the paths of both the free water and bound water out of the fuel bed become more complex, making both evaporation and diffusion more difficult. Therefore, an increase in the fuel bed packing ratio causes a significant increase in time lag (*Jin & Chen, 2012*); Jin & Li, 2011; *Matthews, Gould & McCaw, 2010*). Because higher packing ratios make diffusion more difficult than evaporation, the effect of the fuel bed packing ratio on the diffusion process was greater. The effect of the packing ratio on time lag was the same whether the drying process was calculated using the one-time lag model or the two-time lag model, which may be because the fuel bed packing ratios used in this study were too low and the temperature and humidity ranges were too narrow. This experiment was conducted under high humidity conditions, and a previous study found that the effect of the fuel bed packing ratio on time lag was lower when the humidity was higher (*Zhang, Sun & Jin, 2016*). Therefore, the relationship between key parameters and the fuel bed packing ratio may differ when the packing ratio or humidity are higher than the values in this study.

### Time lag prediction model

The time lag prediction model was established using the fuel bed packing ratio as the independent variable, and the errors were all within the allowable range. This model also revealed the influence of the fuel bed packing ratio on time lag. The prediction effect of $\tau_1$ and $\tau_2$ was better than $\tau$. The sum of the errors of $\tau_1$ and $\tau_2$ were also lower than the error of $\tau$.

## CONCLUSIONS

To more accurately model the drying process of the fuel bed when the initial moisture content exceeds the fiber saturation and analyze the influence of the fuel bed packing ratio on the drying process, this study proposed a two-phase prediction model with the

fiber saturation moisture content as the cut-off point. By analyzing the drying process of different packing ratios of fuel beds, the differences between the one-time lag model and the two-time lag model were compared. The results showed that the two-time lag model is better at modeling the drying process, and that the fuel bed packing ratio has a significant impact on the drying process. Considering the fuel bed packing ratio and distinguishing the two phases of the drying process can effectively improve the prediction accuracy of fuel bed moisture content calculations based on the semi-physical model. This study was only conducted in one temperature and humidity ratio, so it does not cover all meteorological conditions that may occur during the fire risk period in the study area. However, the results of this study provide a foundation for analyzing the effect of the fuel bed packing ratio on its drying process based on the two-time lag model. In addition, the drying mechanism of the fuel bed was different from that of the fuel monomer. When the moisture content of the fuel bed is lower than the saturated moisture content of the fiber, there may still be a certain amount of free water on the surface of the needles in the bed, making evaporation the main drying mechanism. In this study, the fiber saturation point was intentionally used as the cut-off point of the two drying phases, which may cause certain errors in the analysis results. In further research, the moisture content value of the drying mechanism transition, which is important to improving the accuracy of moisture content predictions, should be more accurately defined.

## ACKNOWLEDGEMENTS

We owe our thanks to three anonymous reviewers for their very constructive comments and suggestions.

### Funding

This research was jointly supported by the Science and Technology Support Program of Guizhou Province grant number Qianke Support (2022) General 249, Natural Science Foundation of Guizhou Province grant number ZK (2021) general 158, the Youth Science and Technology Talent Development Project of Education Department in Guizhou Province grant number QJH KY (2021) 251, the China National Natural Science Foundation grant number 31370656. The funders had no role in study design, data collection and analysis, decision to publish, or preparation of the manuscript.

### Grant Disclosures

The following grant information was disclosed by the author:
Science and Technology Support Program of Guizhou Province: (2022), 249.
Natural Science Foundation of Guizhou Province: ZK (2021), 158.
Youth Science and Technology Talent Development Project of Education Department in Guizhou Province: QJH KY (2021), 251.
China National Natural Science Foundation: 31370656.

## Competing Interests

The authors declare there are no competing interests.

## Author Contributions

- Yunlin Zhang conceived and designed the experiments, performed the experiments, analyzed the data, prepared figures and/or tables, authored or reviewed drafts of the article, and approved the final draft.

## Data Availability

The raw measurements are available in the Supplementary File.

## Supplemental Information

Supplemental information for this article can be found online at http://dx.doi.org/10.7717/peerj.14484#supplemental-information.

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
