# Peer review of "Modeling the drying process of Masson pine needle fuel beds under different packing ratios based on two-phase models in the laboratory"

_PeerJ, doi:10.7717/peerj.14484_

## Round 0.1 · original submission · Major Revisions

Dear Yunlin Zhang,

We received 3 evaluations of your paper. All the reviewers provide valuable suggestions for the revision of your paper. Please consider them all and provide detailed answers.

From my side, I strongly recommend that you describe in more detail the experiment, in order to assure its full reproducibility by other researchers. In addition, I warmly recommend that the language will be revised by a professional reviewer.

Last note: e is not the logarithm (a function), but the base of natural logarithms (a number).

I am waiting for the revised version of your manuscript. Please take your time.

Sincerely

Leonardo Montagnani

Reviewer 1 ·

Basic reporting

This manuscript (MS) presents the development of a two-phase model to predict dead fine fuel moisture content (FFMC) of fuel beds of dead pine needles with different packing ratios. Although predicting FFMC is fundamental to fire behaviour modelling, there is a great amount of research dealing with this subject. Thus, new studies on this topic must provide contributions that can unequivocally add to pre-existing work. My opinion is that this manuscript does not meet this criterion, mainly because its lack of clarity makes difficult to understand what was made and how it was made. Namely it was not clear to me how did the field and laboratory experiments where conducted. Thus, I did not find clear how was the data obtained for model development and validation. Below there are some comments that describe just a few examples of what I refer to.

Lns 13-17, This sentence is very long and difficult to read.

Lns 18-23, You mention that you divide the modelling process into two-phases but you do not specify which. There is no information on the experimental data that was used for model development and validation.

Lns 124-129, I did not understand what was the field experiment.

Lns 129-130, I did not understand this sentence.

Lns 139-140, This seems like results rather than methods.

Lns 153-160, Again, I did not quite understand how the indoor experiment was conducted.

Ln 165, You present an equation for your model. Was this equation retrieved from Jin et al. (2000)? Why did you use this model and not other? You say that EMC and time lag can be obtained under different packing ratios. How? Does this mean that each time you run the model you need to pre-obtain these parameters for each packing ratio for a given fuel?

Lns 214-247, You report your results by describing the content of several tables and figures. You should describe you results using figures and tables as a support; you should not describe the content of figures and tables.

Experimental design

no comment

Validity of the findings

no comment

Reviewer 2 ·

Basic reporting

The fuel moisture content determines the occurrence and spread of forest fires. Previous studies only focused on the dynamic change of the fuel moisture content in drought environment, only a few studies focused on the dynamic change of fuel moisture content after extreme high humidity such as rainfall. In particular, the two-phase model combined with the cellulose saturation point makes the study more interesting and valuable. The results also show that the two-phase model is more accurate than the models established by traditional ideas, which is of great significance for the forest fire risk prediction. The study is worthy of publication after minor revision.
1. Introduction- The author has well explained the change mechanism of fuel the moisture content, still needs to insert some references recent years on new research methods and research progress, so as to highlight the advantages of two-phase model;
2. What is the mechanism of packing-ratio in dynamic changes of fuel moisture content;
3. Line105-109- Why masson pine is taken as experimental material?More flammable?
4. Line 110-112- This part is not suitable for Introduction. Move it to Methods;

Experimental design

5. Line114-123- The study is an indoor experiment, not a field experiment, which can mainly describe the basic information of the sample plot, sample time, external conditions and how the sampling was conducted. No need to describe the Study Area.
6. Line 142- How to prepare different packing-ratio fuel bed? When the needles are naturally falled, the packing ratio is 0.016 and the fuel load is 596.06 g/m2 ? Else?

Validity of the findings

7. Line 177 180 183 185…-The unit of fuel moisture content is g·g-1? Why not “%”?
8. line 231 253 291- As shown in line 293, the unit of MRE is “%”.

Additional comments

9. Discussion- Sub-headings are needed to make the readers understand the discuss object in each sections.

Reviewer 3 ·

Basic reporting

The reserach presented in this article, "Modeling the drying process of fuel-beds of Masson
pine needles under different packing ratios," was scientifically interesting and presented in a generally cogent manner. There were some issues with the writing, however, including (but not limited to) tense agreement - such as"has" not "had." Generally speaking, the article should be written in the past tense. The were multiple other instances of awkward, or incorrect syntax as well. I recommend hiring an English language editor. In addition, Italics should be used for all scientific species names.

Experimental design

The encryption of the experimental design was clear an appropriate.

Validity of the findings

The finding of this research were interesting, and useful. There were a couple places where the importance of the research was overstated, however. For example, Line 38, where the author states that "the moisture content of fuels has always been the most important task in forest fire management." Important, perhaps. "Always" the "most" important - no.

Additional comments

Overall, this is a good article. It just needs some editing.

---

## Round 0.2 · accepted · Accept

Dear Dr. Zhang,

I am pleased to inform you that I consider your paper acceptable now.

Sincerely

Leonardo Montagnani

Reviewer 2 ·

Basic reporting

The authors have answered all my questions and made significant changes in the manuscript. I have no further comments on the revised version. I think the current manuscript can be published.

Experimental design

No comment.

Validity of the findings

No comment.

Additional comments

No comment.

Reviewer 3 ·

Basic reporting

Meets standards.

Experimental design

Meets standards.

Validity of the findings

Meets standards.

Additional comments

This article is much improved. I recommend publication in it's current form.